# The Winged Helix Domain of CSB Regulates RNAPII Occupancy at Promoter Proximal Pause Sites

**DOI:** 10.3390/ijms22073379

**Published:** 2021-03-25

**Authors:** Nicole L. Batenburg, Shixin Cui, John R. Walker, Herb E. Schellhorn, Xu-Dong Zhu

**Affiliations:** Department of Biology, McMaster University, Hamilton, ON L8S 4K1, Canada; Batenbn@mcmaster.ca (N.L.B.); cuis19@mcmaster.ca (S.C.); jwalker@mcmaster.ca (J.R.W.); schell@mcmaster.ca (H.E.S.)

**Keywords:** Cockayne syndrome group B (CSB), RNAPII, promoter proximal pause sites, winged helix domain, CSB cancer mutations

## Abstract

Cockayne syndrome group B protein (CSB), a member of the SWI/SNF superfamily, resides in an elongating RNA polymerase II (RNAPII) complex and regulates transcription elongation. CSB contains a C-terminal winged helix domain (WHD) that binds to ubiquitin and plays an important role in DNA repair. However, little is known about the role of the CSB-WHD in transcription regulation. Here, we report that CSB is dependent upon its WHD to regulate RNAPII abundance at promoter proximal pause (PPP) sites of several actively transcribed genes, a key step in the regulation of transcription elongation. We show that two ubiquitin binding-defective mutations in the CSB-WHD, which impair CSB’s ability to promote cell survival in response to treatment with cisplatin, have little impact on its ability to stimulate RNAPII occupancy at PPP sites. In addition, we demonstrate that two cancer-associated CSB mutations, which are located on the opposite side of the CSB-WHD away from its ubiquitin-binding pocket, impair CSB’s ability to promote RNAPII occupancy at PPP sites. Taken together, these results suggest that CSB promotes RNAPII association with PPP sites in a manner requiring the CSB-WHD but independent of its ubiquitin-binding activity. These results further imply that CSB-mediated RNAPII occupancy at PPP sites is mechanistically separable from CSB-mediated repair of cisplatin-induced DNA damage.

## 1. Introduction

The Cockayne syndrome group B protein (CSB), a member of the SWI/SNF2 (switch/sucrose non-fermenting) family, is a multifunctional protein that participates in a wide range of cellular processes. CSB was first described for its role in transcription-coupled nucleotide excision repair (TC-NER) [1,2]. Impaired TC-NER is considered to be an underlying cause of photosensitivity seen in clinical cases of Cockayne syndrome (CS), the majority of which are caused by germline mutations of the *ERCC6* gene encoding CSB [3,4,5]. However, CS patients do not exhibit an increased risk to skin cancer. Aside from photosensitivity, CS is also characterized by severe impairment of physical development, progressive neurological degeneration, cataracts, hearing loss and segmental premature ageing [6]. These phenotypes cannot be fully explained by a deficiency in TC-NER. It has been reported that CSB-deficient cells derived from CS patients exhibit defects in transcription [7,8], oxidative damage repair [9], mitochondria function [10,11], cell division [12], telomere maintenance [13,14] and DNA double-stranded break repair pathway choice [15,16,17], indicative of a role of these additional cellular processes in the development of CS. CSB has also been implicated in transcription regulation [18,19] as well as transcription recovery following genotoxic stress [20,21]. It has been suggested that defects in transcription regulation may underlie some of the neurological features of CS [22,23,24].

CSB resides in an elongating RNA polymerase II (RNAPII) complex [25,26,27]. RNAPII arrested at bulky DNA lesions is thought to serve as the initiating signal for TC-NER [28,29,30,31,32]. It has been suggested that in TC-NER, CSB binds stalled RNAPII and that this binding presents new protein interaction interfaces that facilitate recruitment of downstream repair factors [33]. Recent cryo-EM studies of the structure of the yeast homolog of CSB, Rad26, in complex with stalled/paused RNAPII suggest that although Rad26 does not promote efficient transcriptional bypass of bulky DNA lesions, it promotes paused RNAPII forward translocation on non-damaged templates to stimulate transcription elongation by RNAPII [33]. Promoter proximal pausing of RNAPII represents a key step in regulating transcription elongation at protein-coding genes in metazoans [34]. These findings suggest that CSB may regulate the paused RNAPII at promoter proximal pause (PPP) sites.

CSB contains a N-terminal region, a central ATPase domain and a C-terminal region. The last 76 amino acids of CSB constitute a winged helix domain (WHD) [15,35]. This domain, a protein-protein interaction module, mediates the CSB interaction with several proteins including MRE11/RAD50/NBS1 [16], RIF1 [15] as well as RNAPII [36]. In addition, it has been reported that the WHD of CSB binds ubiquitin [35]. The crystal structure of the CSB-WHD in complex with ubiquitin has revealed that the ubiquitin-binding activity of the CSB-WHD is dependent upon residues in α2 and the C-terminal extremity [35]. Aspartic acid at position 1425 (D1425) and phenylalanine at position 1437 (F1437), both part of α2, make direct interactions with ubiquitin and mutating either D1425 to alanine or F1437 to aspartic acid abrogates binding of the CSB-WHD to ubiquitin [35]. The ubiquitin-binding activity of the WHD has been implicated in efficient TC-NER [36,37], however little is known about the WHD and its ubiquitin-binding activity in transcription regulation. The CSB-WHD is not known to be associated with any CS-causing mutations although the vast majority of CS-causing CSB mutations are nonsense mutations, frameshift mutations and deletions [3]. While much attention has been given to CS-causing CSB mutations over the past decades, recent work suggests that CSB plays a role in cancer prognosis and treatment [38,39,40,41]. Cancer genomics databases of both BioPortal and COSMIC have revealed the presence of cancer-associated mutations in the CSB-WHD albeit their roles have not been characterized.

In this report, we show that CSB is enriched at PPP sites and promotes RNAPII association with PPP sites of several actively transcribed genes through its WHD. We show that the previously reported ubiquitin binding-defective mutations in the CSB-WHD, which impair CSB’s ability to promote cell survival in response to treatment with cisplatin, do not affect its ability to mediate RNAPII association with PPP sites of *ACTB*, *GAPDH* and *RPL13A* genes. In addition, we demonstrate that two cancer-associated mutations, arginine to glutamine at position 1467 (R1467Q) and glycine to arginine at position (G1484R), both of which are located on the opposite side of the CSB-WHD away from its ubiquitin-binding pocket, impair its ability to promote RNAPII occupancy at PPP sites of *ACTB*, *GAPDH* and *RPL13A* genes, providing the first evidence that CSB somatic mutations affect RNAPII association with PPP sites. Taken together, these results suggest that the CSB-WHD mediates RNAPII association with PPP sites in a manner independent of its ubiquitin-binding activity. These results further imply that CSB-mediated RNAPII association with PPP sites is mechanistically separable from CSB-mediated repair of cisplatin-induced DNA damage. 

## 2. Results

### 2.1. CSB Promotes RNAPII Accumulation at PPP Sites of Several Actively Transcribed Genes

Recent cryo-EM studies of the structure of the yeast homolog of CSB, Rad26, in complex with stalled/paused RNAPII suggest that CSB promotes paused RNAPII on non-damaged templates to stimulate transcription elongation [33]. Therefore, we investigated if CSB regulates RNAPII occupancy at PPP sites of seven actively transcribed genes. *ACTB*, *GAPDH*, *RPL13A, CALM3*, *VCL*, *TEFM*, and *MRPL21* are actively transcribed genes that have previously been used for studies of RNAPII occupancy [42]. Using the same primer pair sets previously reported [42] (also shown in Appendix A), we measured RNAPII occupancy at PPP sites, introns and transcription end sites (TES) of these seven genes in human osteosarcoma U2OS cells that are either wild type (WT) or knockout (KO) for CSB [15,16] (Figure 1A). ChIP-qPCR analysis revealed that loss of CSB led to a pronounced reduction in RNAPII occupancy at PPP sites of all seven genes examined (Figure 1B–H). On the other hand, loss of CSB had little impact on RNAPII association with introns and transcription end sites (TES) of all seven genes (Figure 1B–H). To substantiate this finding, we generated U2OS CSB-KO cells stably complemented with either the vector alone or Myc-tagged wild type CSB (Figure 2A) and performed rescue experiments. Myc-CSB is highly overexpressed in U2OS CSB-KO cells compared to endogenous CSB in U2OS parental cells (Figure 2A). ChIP-qPCR analysis revealed that while reintroduction of wild type CSB had little impact on RNAPII association with introns of all seven genes, it rescued RNAPII occupancy at PPP sites of all seven genes in U2OS CSB-KO cells (Figure 2B–H). We also investigated RNAPII occupancy at PPP sites in GM16095, a SV40-transformed CSB-deficient CS cell line [1,13]. We focused on RNAPII occupancy in *ACTB*, *GAPDH* and *RPL13A* genes since these three genes contained a higher level of RNAPII at their PPP sites than the other four genes (*TEFM*, *MRPL21*, *CALM3*, and *VCL)*. ChIP-qPCR analysis showed that overexpression of Myc-CSB in GM16095 cells enhanced RNAPII occupancy at PPP sites of (Appendix A). These results altogether suggest that CSB promotes RNAPII occupancy at PPP sites.

### 2.2. The WHD of CSB Is Necessary for Its Ability to Promote RNAPII Association with PPP Sites

CSB contains a C-terminal WHD [15,35], which has been implicated in its interaction with RNAPII in TC-NER [36]. Therefore, we asked if CSB might be dependent upon its WHD to regulate RNAPII association with PPP sites. To address this question, we generated U2OS CSB-KO cells stably complemented with the vector alone, Myc-CSB or Myc-CSB lacking the WHD (Myc-CSB-∆WHD) (Figure 3A). We chose to focus on RNAPII occupancy in *ACTB*, *GAPDH* and *RPL13A* genes in these complemented cell lines because these three genes contained a higher level of RNAPII at their PPP sites than the other four genes (*TEFM*, *MRPL21*, *CALM3*, and *VCL*) (Figure 1). ChIP-qPCR analysis revealed that while overexpression of Myc-CSB in U2OS CSB-KO cells promoted RNAPII association with PPP sites of all three genes examined, overexpression of Myc-CSB-∆WHD in U2OS CSB-KO cells failed to do so (Figure 3B). The defect of Myc-CSB-∆WHD to promote RNAPII association with PPP sites was unlikely due to a change in its expression since Myc-CSB-∆WHD was expressed at a comparable level to Myc-CSB (Figure 3A). Formally it was possible that a defect in the ability of Myc-CSB-∆WHD to be recruited to PPP sites might have contributed to its deficiency to promote RNAPII association with PPP sites. To address this possibility, we measured CSB occupancy at PPP sites in U2OS CSB-KO cells complemented with the vector alone, Myc-CSB or Myc-CSB-∆WHD. ChIP-qPCR analysis revealed an enrichment of Myc-CSB at PPP sites of all three genes *ACTB*, *GAPDH* and *RPL13A* (Figure 3C), indicating that CSB is associated with PPP sites. Deleting the WHD did not affect Myc-CSB association with any of the three PPP sites (Figure 3C), suggesting that the CSB-WHD is dispensable for CSB association with PPP sites. Taken together, these results suggest that CSB is dependent upon the CSB-WHD to promote RNAPII occupancy at PPP sites of *ACTB*, *GAPDH* and *RPL13A* genes. 

### 2.3. Ubiquitin Binding-Defective Mutations of CSB Sensitize Cells to Cisplatin But Have No Effect on RNAPII Occupancy at PPP Sites

It has been reported that the CSB-WHD binds to ubiquitin and that this binding activity is abrogated by either a D1425A or a F1437D mutation [35]. To investigate if the ubiquitin-binding activity of the CSB-WHD might be required to promote RNAPII occupancy at PPP sites, we generated U2OS CSB-KO cells stably expressing the vector alone, Myc-CSB, Myc-CSB carrying a ubiquitin binding-defective D1425A mutation (Myc-CSB-D1425A) or Myc-CSB carrying a ubiquitin binding-defective F1437D mutation (Myc-CSB-F1437D) (Figure 3D). ChIP-qPCR analysis revealed that both Myc-CSB-D1425A and Myc-CSB-F1437D behaved like Myc-CSB in rescuing RNAPII association with PPP sites of *ACTB*, *GAPDH* and *RPL13A* genes in U2OS CSB-KO cells (Figure 3E). It has been reported that both D1425A and F1437D mutations impair the function of CSB in UV repair [35]. To verify if both D1425A and F1437D mutations affected CSB’s function in DNA repair, we performed clonogenic survival assays. We observed that when overexpressed in U2OS CSB-KO cells, both Myc-CSB-D1425A and Myc-CSB-F1437D were defective in promoting cell survival in response to treatment with cisplatin (Figure 3F), a platinum-based chemotherapeutic drug that is known to induce bulky DNA adducts, in agreement with the previous report [35]. This defect was unlikely due to a change in expression since both Myc-CSB-D1425A and Myc-CSB-F1437D were expressed at a comparable level to Myc-CSB (Figure 3D). Collectively, these results suggest that the ubiquitin-binding activity of CSB is dispensable for RNAPII association with PPP sites of *ACTB*, *GAPDH* and *RPL13A* genes. These results further imply that CSB-mediated RNAPII association with PPP sites is mechanistically separable from CSB-mediated repair of cisplatin-induced DNA damage. 

### 2.4. Cancer-Associated R1467Q and G1484R Mutations Impair CSB’s Ability to Promote RNAPII Occupancy at PPP Sites of ACTB, GAPDH and RPL13A Genes

No CS mutations have been reported to reside in the CSB-WHD, however analysis of cancer genomics databases of both BioPortal and COSMIC revealed the presence of somatic mutations in the CSB-WHD. We have shown that the ubiquitin-binding activity of the CSB-WHD is dispensable for RNAPII occupancy at PPP sites of *ACTB*, *GAPDH* and *RPL13A* genes. To gain further insight into the ubiquitin-binding independent role of CSB-WHD in regulating RNAPII occupancy at PPP sites, we selected five cancer-associated CSB mutations, glutamine to histidine at position 1444 (Q1444H), alanine to serine at position 1445 (A1445S), arginine to leucine at position 1462 (R1462L), arginine to glutamine at position 1467 (R1467Q) and glycine to arginine at position 1484 (G1484R) (Figure 4A), which are located distal to the ubiquitin-binding pocket of the CSB-WHD according to the previously reported crystal structure of the CSB-WHD in complex with ubiquitin [35] (Figure 4B). In addition, we also included in our study the CSB somatic mutation, aspartic acid to asparagine at position 1425 (D1425N), which is from uterine endometrioid carcinoma according to both BioPortal and COSMIC (Figure 4A). We reasoned that the D1425N mutation could serve as a control for a ubiquitin binding-defective somatic mutation since it has been reported that D1425 is directly engaged in binding to ubiquitin [35]. To investigate if any of these six CSB somatic mutations might affect RNAPII occupancy at PPP sites, we generated U2OS CSB-KO cells stably expressing the vector alone, Myc-CSB WT, Myc-CSB-D1425N, Myc-CSB-Q1444H, Myc-CSB-A1445S, Myc-CSB-R1462L, Myc-CSB-R1467Q or Myc-CSB-G1484R. All six CSB mutants were expressed at a comparable level to Myc-CSB (Figure 4C,D). ChIP-qPCR analysis revealed that R1467Q and G1484R mutations but not D1425N, Q1444H, A1445S and R1462L mutations impaired the ability of Myc-CSB to rescue RNAPII occupancy at PPP sites of *ACTB*, *GAPDH* and *RPL13A* genes in U2OS CSB-KO cells (Figure 5A,B). Clonogenic survival assays revealed that all six cancer-associated CSB mutations except for G1484R impaired the ability of Myc-CSB to rescue survival of U2OS CSB-KO cells in response to treatment with cisplatin (Figure 5C–E), suggesting that the observed lack of an effect of D1425N, Q1444H, A1445S and R1462L mutations on RNAPII occupancy at PPP sites is unlikely due to their being silent mutations. Taken together, these results suggest that CSB relies on amino acids R1467 and G1484 but not amino acids D1425, Q1444, A1445 and R1462 to promote RNAPII association with PPP sites. These results also support the notion that CSB-mediated RNAPII association with PPP sites can be mechanistically separated from CSB-mediated repair of cisplatin-induced DNA damage.

## 3. Discussion

It has been suggested that CSB tracks paused RNAPII on non-damage templates to stimulate transcription [33,43], however it has not been previously demonstrated. In this report, we have provided evidence demonstrating that CSB promotes RNAPII association with PPP sites of several actively transcribed genes in cultured cells, supporting the notion that CSB regulates paused RNAPII for transcription elongation. CSB contains a C-terminal winged helix domain (WHD), which binds to ubiquitin [35]. It has been reported that the ubiquitin-binding activity of the CSB-WHD is important for efficient TC-NER [35]. However, several lines of evidence presented here suggest that the ubiquitin-binding activity of the CSB-WHD is dispensable for RNAPII association with PPP sites of *ACTB*, *GAPDH* and *RPL13A* genes. Firstly, both D1425A and F1437D mutations, which have previously been reported to not only abrogate binding of the CSB-WHD to ubiquitin but also sensitize cells to treatment with UV [35], have little effect on the ability of CSB to promote RNAPII occupancy at PPP sites of *ACTB*, *GAPDH* and *RPL13A* genes. Secondly, a cancer-associated D1425N mutation of CSB, which sensitizes cells to cisplatin, does not affect RNAPII occupancy at PPP sites of *ACTB*, *GAPDH* and *RPL13A* genes. D1425 has been reported to be directly engaged in binding to ubiquitin [35]. Thirdly, the two cancer-associated CSB mutations R1467Q and G1484R, located on the opposite side of the CSB-WHD away from its ubiquitin-binding pocket, impair RNAPII association with PPP sites of *ACTB*, *GAPDH* and *RPL13A* genes. Our finding suggests that the surface of the CSB-WHD where R1467 and G1484 reside may play a role in regulating RNAPII association with PPP sites of *ACTB*, *GAPDH* and *RPL13A* genes, which requires further investigation. 

While much attention has been given to CS-causing CSB mutations over the past decades, little is known about CSB somatic mutations that have been identified in cancer cell lines and tumor samples. The work presented here has provided the first evidence that CSB somatic mutations affect CSB function in cultured cells. Increasing evidence suggests that CSB plays a role in cancer prognosis and treatment [38,39,40,41]. Our finding that five of six cancer-associated CSB mutations located in the CSB-WHD, D1425N, Q1444H, A1445S, R1462L and R1467Q, sensitize cells to treatment with cisplatin, a platinum-based chemotherapeutic drug commonly used to treat a number of cancers [44,45,46], suggests that these mutations can serve as biomarkers for cancer diagnosis and treatment. 

CSB is a multifunctional protein that participates in a number of nuclear processes, including transcription regulation [7,8], UV repair [1,2], oxidative DNA damage [9] and DNA DSB repair [15,16,17]. We have previously reported that regulation of CSB function in UV repair is separable from that in DNA DSB repair [37]. Deletion of the first 30 amino acids in the N-terminal region of CSB does not affect its ability to repair UV-induced DNA damage [37] but abrogates its ability to promote cell survival in response to treatment with olaparib, a PARP inhibitor known to be toxic to cells deficient in homologous recombination [15]. In addition, it has been reported that the function of CSB in UV repair is differentially regulated from its role in oxidative DNA damage [47,48]. The work presented in this report suggests that CSB-mediated RNAPII association with PPP sites of *ACTB*, *GAPDH* and *RPL13A* genes is mechanistically separable from CSB-mediated repair of cisplatin-induced DNA damage. Our finding adds to the growing list of evidence that CSB function in various nuclear processes is distinctively regulated from one another. Our finding that five of six cancer-associated CSB mutations, D1425N, Q1444H, A1445S, R1462L and G1484R, differentially affect CSB’s ability to regulate RNAPII association with *ACTB*, *GAPDH* and *RPL13A* genes as well as to promote cell survival in response to cisplatin-induced DNA damage suggests the complexity of the impact of CSB somatic mutations. Future studies are needed to catalog the effect of CSB somatic mutations on its function, which would be expected to aid cancer diagnosis and treatment.

## 4. Materials and Methods

### 4.1. Plasmids and Antibodies

Mammalian expression constructs of Myc-tagged CSB and CSB-ΔWHD have been described [15,17]. Wild type CSB was used as a template to generate, via site-directed mutagenesis, CSB mutants D1425A, F1437D, D1425N, Q1444H, A1445S, R1462L, R1467Q and G1484R, which were cloned into the retroviral expression vector pLPC-NMyc [13]. The primers used to generate these constructs are included in Appendix A. 

Antibodies used include CSB/ERCC6 (A301-345A, Bethyl Laboratories, Montgomery, TX, USA); CSB (A301-347A, Bethyl Laboratories, Montgomery, TX, USA); anti-Myc (9E10, MilliporeSigma, Darmstadt, Germany); RNAPII (05-623, MilliporeSigma, Darmstadt, Germany); γ-tubulin (GTU88, MilliporeSigma, Darmstadt, Germany).

### 4.2. Cell Culture, Transfection, Retroviral Infection and Treatment

All cells were grown in DMEM medium with 10% fetal bovine serum supplemented with non-essential amino acids, L-glutamine, 100 U/mL penicillin and 0.1 mg/mL streptomycin. Cell lines used: Phoenix [13], U2OS [49] (ATCC), U2OS-CSB knockout (KO) [15], GM16095 (Coriell Institute for Medical Research, Camden, NJ, USA). GM16095 is a SV40-transformed cell line derived from GM739 [1]. Cell cultures were routinely fixed, stained with DAPI, and examined for mycoplasma contamination. Retroviral gene delivery was carried out as described [50,51] to generate stable cell lines. DNA transfection was carried out with JetPrime transfection reagent (Polyplus, Illkirch, France) according to the manufacturer’s instructions.

### 4.3. Chromatin Immunoprecipitation (ChIP)

ChIP assays were carried out as described [15]. For each ChIP, 200 μL of the cell lysate was diluted 1:5 in IP dilution buffer [1% Triton X-100, 2 mM EDTA, 20 mM Tris-HCl pH 8.1, 150 mM NaCl]. Out of 1 mL diluted lysate, 20 μL was set aside as input control and the remaining was precleared with protein G sepharose beads (GE Healthcare, Chicago, IL, USA), which were preblocked with BSA and tRNA, and then incubated with primary antibody (1 μg) overnight at 4 °C. The final IP DNA precipitated with ethanol in the presence of 20 μg glycogen (Roche) was resuspended in sterile ddH_2_O. DNA was then analyzed by qPCR using previously described primer pair sets [42] (also shown in Appendix A) and SensiFAST SYBR NO-ROX kit (Bioline, Alvinston, ON, Canada). The threshold cycle (Ct) value of the qPCR reactions for the indicated genes of each ChIP DNA was normalized to that of the input DNA, giving rise to the percentage of input for each ChIP reaction. 

### 4.4. Immunoblotting

Immunoblotting was performed using whole cell extracts as described [52]. Briefly, cell extracts were fractionated by either 6% (for CSB) or 8% (for γ-tubulin) SDS-polyacrylamide gel electrophoresis and then transferred to nitrocellulose membranes. Following immunoblotting, membranes were exposed to Amersham hyperfilms (Cytiva 28906838, Marlborough, MA, USA). Development of the hyperfilm was done on Konica Medical Film Processor SRX-101A, Tokyo, Japan.

### 4.5. Clonogenic Survival Assays

Clonogenic survival assays were done as described [17]. Twenty-four hours post seeding, cells were washed with PBS, treated with varying doses of cisplatin and left to grow in the presence of cisplatin for the entirety of the experiments. Ten days post treatment, colonies were fixed and stained with a solution containing 50% methanol, 7% acetic acid and 0.1% Coomasie blue for 10 min at room temperature. Colonies consisting of more than 32 cells were manually scored on a microscope (Leica EZ4, Concord, ON, Canada).

## Figures and Tables

**Figure 1 ijms-22-03379-f001:**
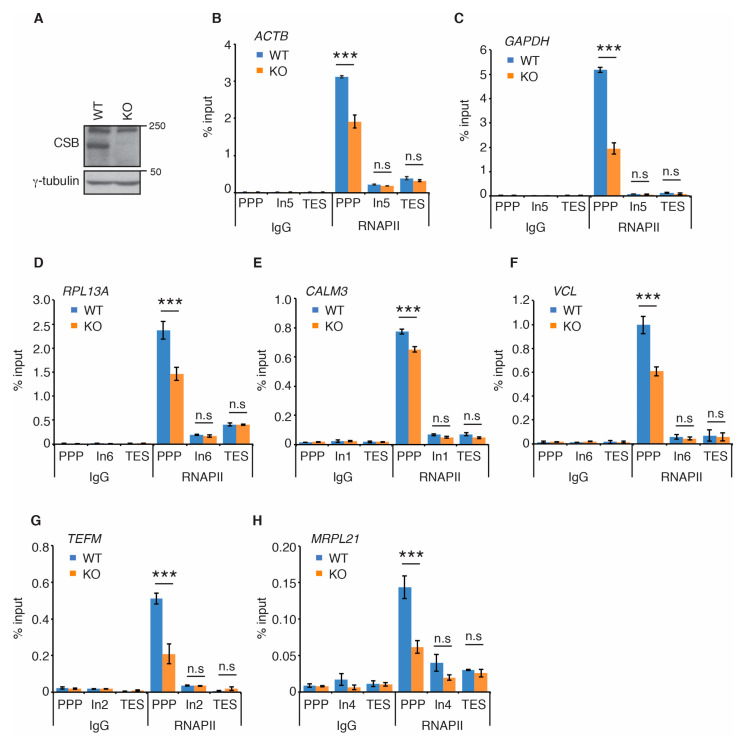
Loss of CSB impairs RNAPII occupancy at PPP sites. (**A**) Western analysis of U2OS CSB-WT and CSB-KO cells. Immunoblotting was performed with anti-CSB and anti-γ-tubulin antibodies. The γ-tubulin blot was used as a loading control in this and subsequent figures. (**B**–**H**) RNAPII ChIP analyses of *ACTB*, *GAPDH*, *RPL13A*, *CALM3*, *VCL*, *TEFM* and *MRPL21* genes in U2OS CSB-WT and CSB-KO cells. Standard errors from three independent experiments are shown. *p* values were derived using the 2-way ANOVA test. *** *p* < 0.001, n.s.: not significant. Abbreviations used in this and subsequent figures—PPP: promoter proximal pause site; ln: intron; TES: transcription end site.

**Figure 2 ijms-22-03379-f002:**
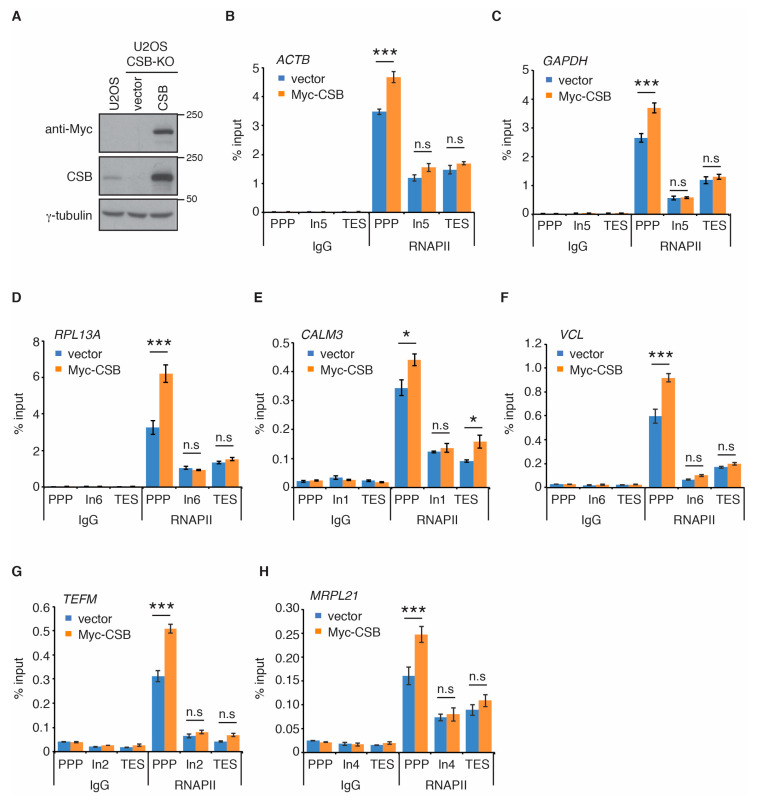
Re-introduction of wild type CSB rescues RNAPII occupancy at PPP sites. (**A**) Western analysis of U2OS cells as well as U2OS CSB-KO cells complemented with the vector alone or Myc-CSB. Immunoblotting was performed with anti-Myc, anti-CSB, and anti-γ-tubulin antibodies. (**B**–**H**) RNAPII ChIP analyses of *ACTB*, *GAPDH*, *RPL13A*, *CALM3*, *VCL*, *TEFM* and *MRPL21* genes in U2OS CSB-KO cells expressing the vector alone or Myc-CSB. Standard errors from three independent experiments are shown. *p* values were derived using the 2-way ANOVA test. * *p* < 0.05; *** *p* < 0.001, n.s.: not significant.

**Figure 3 ijms-22-03379-f003:**
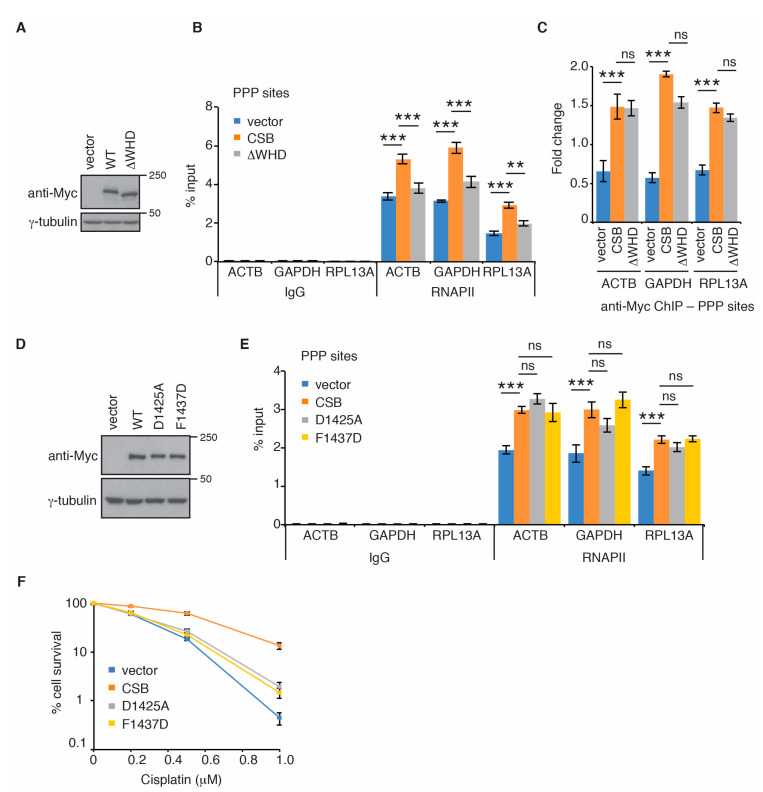
The CSB-WHD mediates RNAPII occupancy at PPP sites in a manner independent of its ability to bind to ubiquitin. (**A**) Western analysis of U2OS CSB-KO cells expressing the vector alone, Myc-CSB WT or Myc-CSB-∆WHD. Immunoblotting was performed with anti-Myc and anti-γ-tubulin antibodies. (**B**) RNAPII ChIP analyses of *ACTB*, *GAPDH* and *RPL13A* genes in U2OS CSB-KO cells complemented with the vector alone, Myc-CSB WT or Myc-CSB-∆WHD. Standard errors from three independent experiments are shown. *p* values were derived using the 2-way ANOVA test. *** *p* < 0.001; ** *p* < 0.01; ns: *p* > 0.05. (**C**) Anti-Myc ChIP analyses of of *ACTB*, *GAPDH* and *RPL13A* genes in U2OS CSB-KO cells complemented with the vector alone, Myc-CSB WT or Myc-CSB-∆WHD. Standard errors from three independent experiments are shown. *p* values were derived using the 2-way ANOVA test. *** *p* < 0.001; ns: *p* > 0.05. (**D**) Western analysis of U2OS CSB-KO cells expressing the vector alone, Myc-CSB WT, Myc-CSB-D1425A or Myc-CSB-F1437D. Immunoblotting was performed with anti-Myc and anti-γ-tubulin antibodies. (**E**) RNAPII ChIP analyses of *ACTB*, *GAPDH* and *RPL13A* genes in U2OS CSB-KO cells complemented with the vector alone, Myc-CSB WT, Myc-CSB-D1425A or Myc-CSB-F1437D. Standard errors from three independent experiments are shown. *p* values were derived using the 2-way ANOVA test. *** *p* < 0.001; ns: *p* > 0.05. (**F**) Cisplatin clonogenic survival assays of U2OS CSB-KO cells complemented with the vector alone, Myc-CSB WT, Myc-CSB-D1425A or Myc-CSB-F1437D. Standard deviations from three independent experiments are indicated.

**Figure 4 ijms-22-03379-f004:**
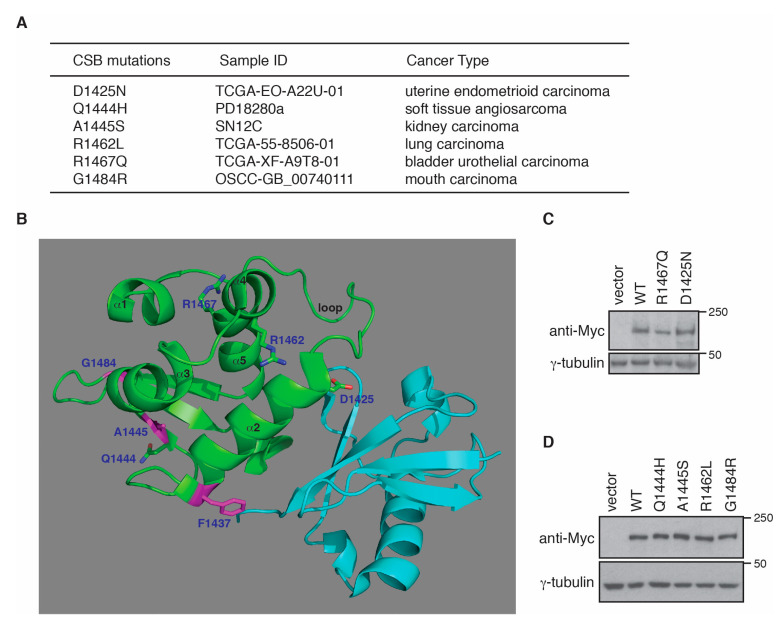
Cancer-associated CSB mutations in the CSB-WHD have little effect on CSB expression. (**A**) Six cancer-associated CSB mutations taken from both BioPortal and COSMIC databases. (**B**) Cartoon representation of the structure of the CSB-WHD in complex with ubiquitin (PDB code 6A6I). The CSB-WHD is coloured green, and ubiquitin is coloured cyan. Sidechains for D1425, F1437, Q1444, A1445, R1462, R1467 and G1484 are shown in stick representation, with nitrogen and oxygen atoms coloured in blue and red, respectively. Sidechains of F1437 and A1445, along with the position of G1484 on the cartoon backbone, are coloured magenta. For the CSB-WHD, α helices are numbered, and the N-terminal loop is identified. The figure was generated using PyMOL (www.pymol.org, accessed on 10 February 2021). (**C**) Western analysis of U2OS CSB-KO cells expressing the vector alone, Myc-CSB WT, Myc-CSB-R1467Q or Myc-CSB-D1425N. Immunoblotting was performed with anti-Myc and anti-γ-tubulin antibodies. (**D**) Western analysis of U2OS CSB-KO cells expressing the vector alone, Myc-CSB WT, Myc-CSB-Q1444H, Myc-CSB-A1445S, Myc-CSB-R1462L or Myc-CSB-G1484R. Immunoblotting was performed with anti-Myc and anti-γ-tubulin antibodies.

**Figure 5 ijms-22-03379-f005:**
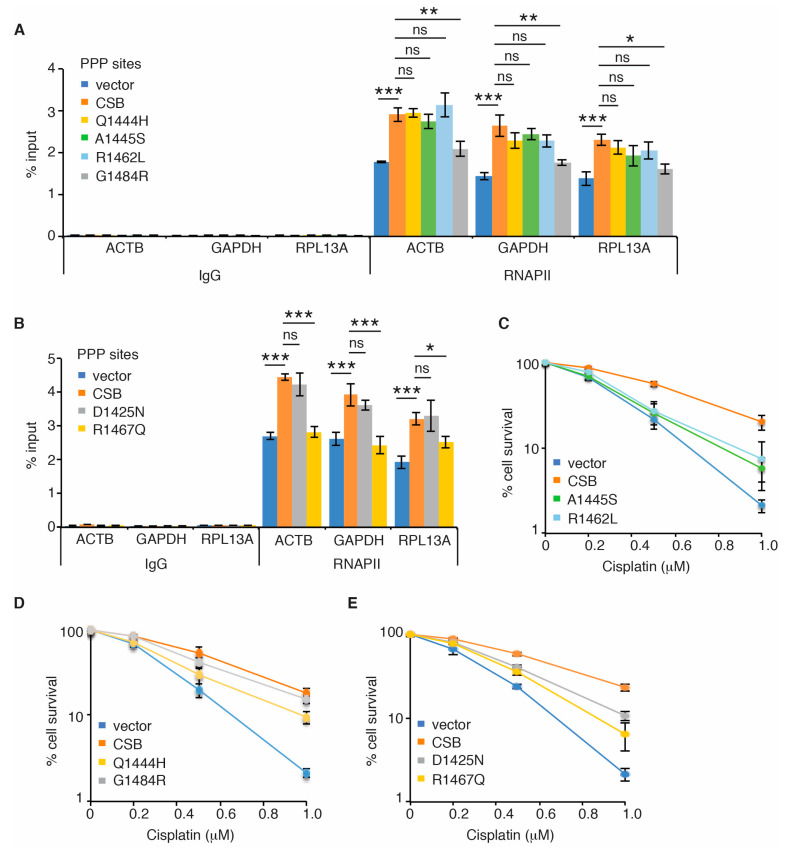
Cancer-associated R1467Q and G1484R mutations of CSB impair RNAPII pausing at PPP sites. (**A**) RNAPII ChIP analyses of *ACTB*, *GAPDH* and *RPL13A* genes in U2OS CSB-KO cells expressing the vector alone, Myc-CSB WT, Myc-CSB-Q1444H, Myc-CSB-R1462L, Myc-CSB-R1467Q or Myc-CSB-G1484R. Standard errors from three independent experiments are shown. *p* values were derived using the 2-way ANOVA test. * *p* < 0.05; ** *p* < 0.01; *** *p* < 0.001; ns: *p* > 0.05. (**B**) RNAPII ChIP analyses of *ACTB*, *GAPDH* and *RPL13A* genes in U2OS CSB-KO cells expressing the vector alone, Myc-CSB WT, Myc-CSB-D1425N or Myc-CSB-R1467Q. Standard errors from three independent experiments are shown. *p* values were derived using the 2-way ANOVA test. * *p* < 0.05; ** *p* < 0.01; *** *p* < 0.001; ns: *p* > 0.05. (**C**) Cisplatin clonogenic survival assays of U2OS CSB-KO cells expressing the vector alone, Myc-CSB WT, Myc-CSB-A1445S or Myc-CSB-R1462L. Standard deviations from three independent experiments are indicated. (**D**) Cisplatin clonogenic survival assays of U2OS CSB-KO cells expressing the vector alone, Myc-CSB WT, Myc-CSB-Q1444H or Myc-CSB-G1484R. Standard deviations from three independent experiments are indicated. (**E**) Cisplatin clonogenic survival assays of U2OS CSB-KO cells expressing the vector alone, Myc-CSB WT, Myc-CSB-D1425N or Myc-CSB-R1467Q. Standard deviations from three independent experiments are indicated.

## Data Availability

All data are contained within this manuscript.

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
