# Peer review of "The Winged Helix Domain of CSB Regulates RNAPII Occupancy at Promoter Proximal Pause Sites"

_ijms, 2021, doi:10.3390/ijms22073379_

Round 1

Reviewer 1 Report

Review of the manuscript by Batenburg et al., titled “the winged helix domain of CSB regulates RNAPII occupancy at promotor proximal pausing sites”. In this study the authors investigated by Chromatin-Immunoprecipitation the influence of the winged-helix domain (WHD) and mutation therein on promoter-proximal pausing (PPP) of RNA polymerase II. Monitoring PPP, introns and transcription end sites the authors conclusively show, that the WHD domain of CSB independent of its ubiquitin binding activity regulates PPP of RNA polymerase II and cancer-associated CSB mutations impair PPP.  These works clarify the role of CSB in PPP, a step regulating transcription elongation.

The manuscript is structured in a very appealing manner, the experiments are presented in a logical flow and the description in the result part appropriate. The discussion is well written and highlights the central findings.

Major points: It would be complement the study when the authors would show a qPCR in Figure 1 quantifying the mRNA-expression of the depicted genes after CSB knockdown

Minor points: Legend Figure 2 A describes western blots of U2OS cells as well as U2OS CSB-KO cells, but only the latter are shown.

No more criticism, a very nice study that was a pleasure to read!

Author Response

Point-by point response to Reviewer 1:

Point #1: “It would be complement the study when the authors would show a qPCR in Figure 1 quantifying the mRNA-expression of the depicted genes after CSB knockdown.”

Response: Following Reviewer 1’s suggestion, we performed RT-qPCR experiments to measure mRNA levels of seven genes shown in Figure 1. We did not detect any significant change in the mRNA levels of these seven genes between U2OS CSB-WT and CSB-KO cells. We would like to point out that although loss of CSB impairs RNAPII occupancy at PPP sites, it does not affect RNAPII occupancy within the gene bodies (e.g. introns and TESs) of the seven genes. Perhaps, the impaired RNAPII occupancy at PPP sites alone in U2OS CSB-KO cells is insufficient to cause a significant change in the levels of steady-state mRNAs of these seven genes. Alternatively, RT-qPCR may not be sensitive enough to detect a change in mRNA levels resulting from impaired RNAPII occupancy at PPP sites alone. Future studies (e.g. RNA-seq analysis) will be needed to further investigate the effect of impaired RNAPII occupancy on mRNA levels in CSB-KO cells. Nevertheless, our finding that CSB regulates RNAPII occupancy at PPP sites agrees with previous reports that CSB plays a role in transcription regulation.

Point #2: “Legend Figure 2 A describes western blots of U2OS cells as well as U2OS CSB-KO cells, but only the latter are shown.”

Response: We have replaced the original Figure 2A with a new Figure 2A, in which U2OS parental cells were included in the western blot.

Reviewer 2 Report

In this manuscript, Batenburg et al. have investigated the importance of the CSB winged helix domain (WHD) on RNAPII occupancy at PPP sites. They demonstrate that a CSB mutant lacking the WHD domain is unable to promote RNAPII localisation to these sites and suggest this is due to a role of the WHD domain that is separate from its characterised ubiquitin-binding function.

I find the data convincing and the article to be well-written.

I do though have the following specific comments:

Comments:

  • The authors have generated Myc-CSB-expressing cell lines in the U2OS CSB-KO background. How similar were the expression levels of Myc-CSB in these cells, compared to CSB expression in parental U2OS cells? Could the authors provide a gel to demonstrate this?
  • One possible explanation for the author’s findings in Figure 5, is that the R1467Q and G1484R mutations of CSB disrupt the association with RNAPII. Have the authors tested this? Immunoprecipitations between WT, R1467Q and G1484R CSB with RNAPII would be a useful addition to the paper.
  • It is interesting that all six cancer mutations in figure 5 impaired the ability of Myc-CSB to rescue survival of U2OS CSB-KO cells in response to treatment with cisplatin. This is even though these mutations were examined to 'gain further insight into the ubiquitin-binding independent role of CSB-WHD'. Although these mutations are located away from the ubiquitin-binding pocket, it would be useful to know if they do actually impair ubiquitin-binding, perhaps through disruption of the WHD structure. If they do not, this might suggest the CSB-WHD domain also has a role in the repair of cisplatin-induced DNA damage that is independent of ubiquitin-binding.
  • The immunoblots in Figures 1, 2, and 3A have a grey background, while those in figures 3D and 4 have more of a grey-brown background. Were these images prepared in different ways? Although it does not affect the interpretation of the data, the authors might wish to double check this for consistency.
  • The term in vivo is not really appropriate to describe experiments performed in cell lines that were grown under in vitro I suggest the authors instead use terms such as ‘in cultured cells’, or ‘in cells lines’.
  • Line 302. I do not believe it is adequate to state “The primers used to generate these constructs are available upon request”. Please include the primers used for SDM in the paper, perhaps in the supplementary methods with the other primers.
  • Line 310: How were cells examined for mycoplasma?
  • In Line 327, the authors state: ‘Immunoblotting was performed as described [15]’. In the cited paper - a previous publication by the authors - the materials and methods state: ‘…immunoblotting were performed as described (Wu et al, 2007a; McKerlie et al, 2012)’. Wu et al, 2007 states ‘Immunoblotting and immunofluorescence were done using standard protocols, as described10”’. McKerlie et al 2012 states ‘Protein extracts and immunoblotting were carried out essentially as described (27,34)’. After going back this far, I still do not know how the authors performed their immunoblotting experiments e.g. what type of gels they used, which type of membrane was transferred to, or which imaging system was used. Please add these details.
  • In Line 333-334 of the materials and methods, under ‘Clonogenic survival assays’, the authors write ‘Colonies consisting of more than 32 cells were scored’. That is very specific – how did the authors manage this?

Author Response

Point-by point response to Reviewer 2:

Point #1: “The authors have generated Myc-CSB-expressing cell lines in the U2OS CSB-KO background. How similar were the expression levels of Myc-CSB in these cells, compared to CSB expression in parental U2OS cells? Could the authors provide a gel to demonstrate this?”

Response: We have performed new experiments as suggested to evaluate the expression levels of Myc-CSB vs endogenous CSB. Our data show that Myc-CSB in U2OS CSB-KO cells is highly overexpressed compared to endogenous CSB in U2OS cells (Fig. 2A). We have included this new data in Fig. 2A in the revised manuscript.

Point #2: “One possible explanation for the author’s findings in Figure 5, is that the R1467Q and G1484R mutations of CSB disrupt the association with RNAPII. Have the authors tested this? Immunoprecipitations between WT, R1467Q and G1484R CSB with RNAPII would be a useful addition to the paper.”

Response: It has been suggested that the WHD of CSB mediates the interaction of CSB with RNAPII (Sin et al 2016), however, we have not been able to reproduce this finding. Under our experimental conditions, deletion of the WHD where R1467Q and G1484R reside does not affect CSB’s ability to pull down RNAPII. Recently, cryo-EM studies of the structure of the yeast homolog of CSB, Rad26, in complex with stalled/paused RNAPII have shown that the ATPase domain of CSB makes extensive contacts with RNAPII (Xu et al 2017). Whether and how the CSB-WHD regulates the interaction of CSB with RNAPII will require future investigation.

Point #3: “It is interesting that all six cancer mutations in figure 5 impaired the ability of Myc-CSB to rescue survival of U2OS CSB-KO cells in response to treatment with cisplatin. This is even though these mutations were examined to 'gain further insight into the ubiquitin-binding independent role of CSB-WHD'. Although these mutations are located away from the ubiquitin-binding pocket, it would be useful to know if they do actually impair ubiquitin-binding, perhaps through disruption of the WHD structure. If they do not, this might suggest the CSB-WHD domain also has a role in the repair of cisplatin-induced DNA damage that is independent of ubiquitin-binding.

Response: We would like to point out that five out of six (not all six) cancer mutations in Figure 5 impaired the ability of Myc-CSB to rescue survival of U2OS CSB-KO cells in response to treatment with cisplatin. The G1484R mutation has no effect on CSB’s ability to rescue survival of U2OS CSB-KO cells in response to treatment with cisplatin.

We appreciate that this reviewer raises an interesting question as to whether the CSB-WHD also has a role in the repair of cisplatin-induced DNA damage independently of its ubiquitin-binding. However, we feel that addressing this question is beyond the scope of the current manuscript since the current manuscript focuses on the role of the CSB-WHD in RNAPII occupancy at PPP sites in the absence of DNA damage.

Point #4: “The immunoblots in Figures 1, 2, and 3A have a grey background, while those in figures 3D and 4 have more of a grey-brown background. Were these images prepared in different ways? Although it does not affect the interpretation of the data, the authors might wish to double check this for consistency.”

Response: The images of the immunoblots in Figures 1, 2 and 3A were prepared in the “greyscale” mode whereas the images of the immunoblots in Figures 3D and 4 were prepared in the “RGB” mode. We have re-prepared Figures 3D and 4 in “greyscale” mode in the revised manuscript.

Point #5: “The term in vivo is not really appropriate to describe experiments performed in cell lines that were grown under in vitro I suggest the authors instead use terms such as ‘in cultured cells’, or ‘in cells lines’.”

Response: We have deleted the phrase “in vivo” and instead used the phrase “in cultured cells” on page 9. In addition, we have also replaced the phrase “in vivo” with the phrase “in cultured cells” on page 10 in the revised manuscript.

Point #6: “Line 302. I do not believe it is adequate to state “The primers used to generate these constructs are available upon request”. Please include the primers used for SDM in the paper, perhaps in the supplementary methods with the other primers.”

Response: We have included the primers used to generate CSB mutations in the Supplementary Table S2 in the revised manuscript.

Point #7: “Line 310: How were cells examined for mycoplasma?”

Response: Cells were examined for mycoplasma through DAPI staining and visualization on microscope. We have included this information in the revised manuscript.

Point #8: “In Line 327, the authors state: ‘Immunoblotting was performed as described [15]’. In the cited paper - a previous publication by the authors - the materials and methods state: ‘…immunoblotting were performed as described (Wu et al, 2007a; McKerlie et al, 2012)’. Wu et al, 2007 states ‘Immunoblotting and immunofluorescence were done using standard protocols, as described10”’. McKerlie et al 2012 states ‘Protein extracts and immunoblotting were carried out essentially as described (27,34)’. After going back this far, I still do not know how the authors performed their immunoblotting experiments e.g. what type of gels they used, which type of membrane was transferred to, or which imaging system was used. Please add these details.”

Response: We have replaced the reference [15] with the reference [50], which describes the preparation of whole cell extracts and immunoblotting. In addition, we have added the requested information regarding the type of gels, the type of membrane as well as the film processor in the revised manuscript.

Point #9: “In Line 333-334 of the materials and methods, under ‘Clonogenic survival assays’, the authors write ‘Colonies consisting of more than 32 cells were scored’. That is very specific – how did the authors manage this?”

Response: Colonies were manually scored on a microscope (Leica EZ4). We have included this statement in the revised manuscript.

Reviewer 3 Report

In this manuscript, Batenburg et al. provide a detailed functional characterization of the C-terminal winged helix domain (WHD) of the Cockayne syndrome group B protein CSB in relation to its role in transcription. They show that this domain regulates RNAPII abundance at promoter proximal pause (PPP) sites of a number of actively transcribed genes. In addition, by mutating CSB-WHD at specific sites, they demonstrate that CSB role in promoting RNAPII association at PPP sites is mechanistically separable from its role in DNA damage repair.

The manuscript is clearly written and experiments are correctly conducted, so I believe this work is worthy of publication in International Journal of Molecular Sciences. I only have one major suggestion. All functional experiments have been conducted in U2OS cells knocked out for CSB, it could improve the paper if the authors could replicate observations on PPP sites in a CSB-lacking cell line obtained from a CS patient, even better in patient primary cells.

Minor comments:

Introduction

  1. I would stress the cancer-free nature of the hereditary disorder CS.
  2. More recent references for the disease and related mutations should be included (e.g. Karikkineth et al. Ageing Res Rev 2017, Calmels et al. J Med Genet 2018).
  3. For clarity, full names of the mutated residues should be reported when first cited.

Figures

Please add missing statistics in Fig 1 and Fig 2

Supplementary Table

Please add Reference sequences and primer positions.

Typing errors

Line 81: independently of, please correct to independent of

Line 248: transcrpition

Author Response

Point-by point response to Reviewer 3:

Point #1: “… I only have one major suggestion. All functional experiments have been conducted in U2OS cells knocked out for CSB, it could improve the paper if the authors could replicate observations on PPP sites in a CSB-lacking cell line obtained from a CS patient, even better in patient primary cells.”

Response: We have performed new RNAPII ChIP-qPCR experiments in GM16095, a SV4-transformed CS cell line. We focused on RNAPII occupancy in ACTB, GAPDH and RPL13A genes since these three genes contained a higher level of RNAPII at their PPP sites than the other four genes (TEFM, MRPL21, CALM3, and VCL). Our new data show that overexpression of wild type CSB in GM16095 cells leads to an increase in RNAPII occupancy at PPP sites of ACTB, GAPDH and RPL13A genes, supporting the notion that CSB promotes RNAPII occupancy at PPP sites. We have included this new data as Supplementary Figure S1 in the revised manuscript.

Point #2: “I would stress the cancer-free nature of the hereditary disorder CS.”

Response: We have included “CS patients do not exhibit increased risk to skin cancer” in the introduction.

Point #3: “More recent references for the disease and related mutations should be included (e.g. Karikkineth et al. Ageing Res Rev 2017, Calmels et al. J Med Genet 2018).”

Response: We have included these recent references as references [5] and [6] in the introduction.

Point #4: “For clarity, full names of the mutated residues should be reported when first cited.”

Response: We have described full names of the mutated residues as suggested in the revised manuscript.

Point #5: “Please add missing statistics in Fig 1 and Fig 2.”

Response: We have added missing statistics in Figures 1 and 2 in the revised manuscript.

Point #6: “Please add Reference sequences and primer positions.”

Response: We adopted the primer pair sets that are reported by Shivji et al. (2018). We have included this reference in the main text as well as in the Supplementary Table S1 in the revised manuscript.

Point #7: “Line 81: independently of, please correct to independent of”

Response: It has been corrected.

Point #8: “Line 248: transcrpition”

Response: It has been corrected.

Round 2

Reviewer 2 Report

The authors have addressed my concerns. I recommend the manuscript be accepted in present form.